# A Review of Histocytological Events and Molecular Mechanisms Involved in Intestine Regeneration in Holothurians

**DOI:** 10.3390/biology11081095

**Published:** 2022-07-22

**Authors:** Fang Su, Hongsheng Yang, Lina Sun

**Affiliations:** 1CAS Key Laboratory of Marine Ecology and Environmental Sciences, Institute of Oceanology, Chinese Academy of Sciences, Qingdao 266071, China; sufang19@mails.ucas.ac.cn (F.S.); hshyang@qdio.ac.cn (H.Y.); 2Laboratory for Marine Ecology and Environmental Science, Qingdao National Laboratory for Marine Science and Technology, Qingdao 266237, China; 3Center for Ocean Mega-Science, Chinese Academy of Sciences, Qingdao 266071, China; 4CAS Engineering Laboratory for Marine Ranching, Institute of Oceanology, Chinese Academy of Sciences, Qingdao 266071, China; 5University of Chinese Academy of Sciences, Beijing 100049, China; 6Shandong Province Key Laboratory of Experimental Marine Biology, Qingdao 266071, China; 7The Innovation of Seed Design, Chinese Academy of Sciences, Wuhan 430071, China

**Keywords:** sea cucumber, evisceration, regeneration, holothurian, echinoderm

## Abstract

**Simple Summary:**

Many species of sea cucumber in Echinodermata may eviscerate most of or even all internal organs when encountering predators or adverse environments, and can achieve regeneration within a certain time. Although regeneration time varies, the mechanism is common. This paper reviewed the intestinal regeneration process of sea cucumbers from the perspectives of histocytology and molecular mechanism. Echinodermata has a special evolutionary position between achordate and chordate, so we hope to explore the common regeneration conserved signals between invertebrates and vertebrates by recording the intestinal regeneration of sea cucumbers.

**Abstract:**

Most species of the class Holothuroidea are able to regenerate most of their internal organs following a typical evisceration process, which is a unique mechanism that allows sea cucumbers to survive in adverse environments. In this review, we compare autotomy among different type of sea cucumber and summarize the histocytological events that occur during the five stages of intestinal regeneration. Multiple cytological activities, such as apoptosis and dedifferentiation, take place during wound healing and anlage formation. Many studies have focused on the molecular regulation mechanisms that underlie regeneration, and herein we describe the techniques that have been used as well as the development-related signaling pathways and key genes that are significantly expressed during intestinal regeneration. Future analyses of the underlying mechanisms responsible for intestinal regeneration should include mapping at the single-cell level. Studies of visceral regeneration in echinoderms provide a unique perspective for understanding whole-body regeneration or appendage regeneration.

## 1. Introduction

Regeneration, defined as the replacement of damaged or lost parts following injury without scarring or loss of functionality, is a widespread phenomenon in almost all Metazoans [1,2]. For example, planarians have a rich library of adult stem cells, and even when cut into 279 parts, they were still able to regenerate a complete individual [3]. Hydra stem cells also have unlimited self-renewal ability [4]. Most vertebrates can non-uniformly regenerate some tissues or organs after injury [5]. Teleosts have a strong ability to regenerate various tissues and organs, including scales, muscles, fins, spinal cords, and hearts [6]. Urodele amphibians, such as newts and axolotls, can regenerate jaws, tails, limbs, and lenses [7,8,9], and reptiles can re-form appendages [10]. Mammals rely on undifferentiated adult stem cells or dedifferentiation of fully differentiated cells for local regeneration [11]. As the evolutionary status of organisms increases, the regeneration potential is gradually limited.

All classes of echinoderms (Echinoidea (sea urchins and sand dollars), Ophiuroidea (brittle stars), Asteroidea (sea stars), Crinoidea (sea lilies and feather stars), and Holothurioidea (sea cucumbers)) have the capacity to regenerate missing body parts [12]. Sea stars are well-known for regenerating arms [13]. Huet et al. [14,15] studied the ultrastructure and development of the neuroepithelial cells as well as the role of the nervous system during arm regeneration. In addition to adult appendage regeneration, sea star larvae can rapidly and completely regenerate to form fully functional individuals after bisection [16]. Arm regeneration is also a common phenomenon in crinoids and ophiuroids, both of which have long and fragile arms that can be lost to predation or physical stress [12,17]. In addition to arms, they can regenerate internal organs such as the pyloric caeca, cardiac stomach, gonads, and gut [12]. Compared to the other groups, the need for regeneration of sea urchins is reduced and they can regenerate external appendages (spines, pedicellariae), test, and tube feet [12,18]. In contrast, sea cucumbers can regenerate almost all of their viscera.

In 1930, the Italian scholar Bertolini studied the histological characteristics of the regenerating sea cucumber digestive tract and identified various tissues involved in the regeneration process for the first time [19] and published the research results [20]. The pattern of evisceration and regeneration and the duration of regeneration vary among sea cucumber species. For example, in *Apostichopus japonicus*, the newborn intestine recovers digestive function in 15 days and returns to the normal physiological state in 45 days [21]. The digestive tract of *Holothuria scabra* regrows within 1 week [22], whereas the new intestine of *Holothuria glaberrima* returns to its original shape ~1 month after evisceration [23].

In this review, we outline the structure of the digestive tract of sea cucumbers, describe the mode and process of evisceration, assess research methodologies, and progress in our understanding of intestine regeneration at the histological, cytological, and molecular levels.

## 2. Autotomy and Evisceration

Autotomy (self-cutting) refers to the adaptive detachment of animal body parts as the body’s self-defense mechanism, and it is regulated by nerves [24,25,26]. Many sea cucumber species of the orders Holothuriida, Synallactida, and Dendrochirotida are capable of eviscerating their intestine in response to stressful stimuli, such as crowding, excessive ambient temperature, hypoxia, or encountering predators [27,28]. However, strong stimuli do not always induce autotomy [29]. In addition, some species, such as *Eupentacta quinquesemita* [30] and *Parastichopus californicus* [31], undergo seasonal evisceration. During evisceration, holothurians detach most of the viscera (e.g., intestine, respiratory trees, gonads) automatically under the control of nerves [32] and then eject them from the coelom. Subsequently, the tissues regrow rapidly (Table 1).

Some dendrochirotids, such as *Cladolabes schmeltzii* and *Colochirus robustus*, lack the ability to eviscerate [39]. Almost all regenerative species of dendrochirotida eviscerate anteriorly. However, *Pseudocolochirus violaceus* [39] eviscerates through the cloaca. Holothuriida and synallactida generally eviscerate through the cloaca, although some eviscerate by tearing the body wall [39]. Species of *Holothuria* retain a single respiratory tree in vivo, on either the left or right side [19]. Whether the gonads are eviscerated depends on the reproductive stage.

Evisceration in sea cucumbers occurs either from the rupture of the anterior end of the body (anterior) or the cloaca (posterior) [26,33]. Anterior evisceration, which is most common in Dentrochirotida, results in the loss of almost all of the viscera, including the feeding tentacles, aquapharyngeal bulb, the whole digestive tract and attached haemal vessels, a respiratory tree, and possibly the gonads, but the cloaca remains. These structures are ejected from the rupture of the anterior end of the body leading to a hole (wound) [33]. In posterior evisceration, which mainly occurs in the holothuriids and synallactids, the digestive tract between the oesophagus and the cloaca is lost, along with attached haemal vessels, one or two respiratory trees, and sometimes the gonads. These structures are ejected through the cloaca, although in a few species it occurs directly through the body wall (e.g., *Stichopus chloronotus* [39], *Holothuria difficilis*, *H. surinamensis*, and *H. parvula*) [19].

During the posterior evisceration process, coelomic fluid is ejected from the cloaca, followed by acute contractions of the body wall. The pharyngeal retractor expands, and the tentacles and tube feet swing rapidly and irregularly [40]. Species that undergo anterior evisceration exhibit rapid softening of the ligaments and tendons as well as rupture of the introvert [19,24,35]. After evisceration, the body wall slowly returns to the relaxed state. 

The function of the collagenous tissues in the sea cucumber body is to switch between rigid and flexible states to maintain rapid changes in mechanical properties, but irreversible disintegration of the collagenous tissues occurs during autotomy [24,25,26]. At the fracture plane, the most abundant soft tissue with mechanical significance is mutable collagenous tissue (MCT). MCT exists in the form of dermal connective tissue, interossicular ligaments, and tendons, and it is involved in the autotomy mechanism in all echinoderms [41]. A small fraction of MCTs contain myocytes, and all MCTs contain cells rich in inhomogeneous vacuoles, which seems to be lysosomal [41]. Additionally, juxtaligamental cells, which contain electron-dense granules, are effector cells that directly change the tensile properties of MCT [41]. The detachment mechanism of the evisceration process depends on the rapid, irreversible loss of tensile strength of MCT at the autotomy plane [26]. Changes in MCT tensility do not involve changes in the mechanical properties of hollow microfibrils or collagen fibrils, and the mutability depends on changes in the cohesion that holds the fibrils together. Juxtaligamental granules in the dermis of sea cucumbers contain chemical factors that affect interfibrillar cohesion and are sensitive to changes in extracellular calcium ion concentration. Calcium ion concentration is the direct cause of MCT interfiber cohesion, and the stiffness of MCT becomes stronger when the concentration of Ca^2+^ increases [41]. Although juxtaligamental cells are not involved in intestinal autotomy, neurosecretory-like processes containing large dense vesicles scattered through the connective tissue control MCT mutability [24]. MCTs also are permeated by granule-containing processes belonging to neurosecretory-like perikarya, which may be in synaptic contact with motor neurons [25]. 

The release of the neurosecretory evisceration factor (EF) into the coelomic fluid results in a complete loss of tension in the connective tissue matrix [25,32], triggering autotomy, and muscle contraction facilitates the separation of the autotomy structures [24]. However, autotomy may occur during both the relaxation and the contraction phase of a cycle rather than at maximum tension [42]. Neither the radial nerve cord nor the nerve ring is an abundant source of EF; instead, it is found primarily in the axonal tracts of the haemal system [42]. Peritoneum is another major source of EF [24]. EF is a multifunctional molecule [42] that also exists in non-eviscerating species [32]. It may be a neuropeptide, but little is known about its chemical properties and whether it consists of one or several molecules [24]. Although the *A. japonicus* genome is available, it is still unclear what the EF is. Mechanical, chemical, and hypoxic stress can all lead to evisceration [43], which is a complex physiological process under the control of nerves. Before and after evisceration, genes related to muscle contraction and neurotransmitter secretion are differentially expressed [40], and metabolites related to oxidative stress and nerves change significantly [44].

In the laboratory, evisceration induction usually involves intraperitoneal injection of potassium chloride (KCl) solution at a concentration of 0.35 mol/L (lower than the osmolality of coelomic fluid (0.54 mol/L)) or occasionally distilled water [42]. The potassium ion is a nerve active substance, and injection of KCl solution into the coelom stimulates the release of EF and acetylcholine. The combined effect of the two leads to decreased MCT viscosity and muscle contraction, ultimately leading to tissue breakdown [32,41,42].

Many studies have explored the histological and physiological changes that occur during holothurian evisceration and regeneration [39,45,46,47], but few studies have focused on the molecular mechanism responsible for evisceration. Using Illumina sequencing, Ding et al. [40] investigated the genes that are differentially expressed during the evisceration process of *A. japonicus* and identified genes involved in muscle contraction, hormone and neurotransmitter secretion, nerve and muscle damage, energy support, cellular stress, and apoptosis.

## 3. Histology and Cytology of Intestinal Regeneration

Understanding the histological and cytological changes that occur during various stages of regeneration of the holothurian digestive tract requires knowledge of the structure of the normal digestive tract (Figure 1A). From the mouth to the cloaca, the digestive tract is roughly divided into the oral complex, pharynx, esophagus, stomach (although some species, such as *H. glaberimma,* do not have a stomach [48]), intestine, and cloaca. In some species, the size and structure of each region is determined by feeding behavior and digestive tract physiology, so it is species-specific.

Studies of the histology and cytology of the intestinal wall of sea cucumbers indicate slight differences in tissue layers and cellular composition among species or in different regions of the digestive tract [49,50,51], but the structural composition is similar. Tissues of the intestinal wall from outside to inside are as follows: coelomic epithelium exposed to the coelom, outer connective tissue, longitudinal muscles, circular muscles, inner connective tissue, and luminal epithelium, which is an endodermal derivative [52] that is exposed to the intestinal lumen [19,48,53]. The luminal epithelium is monolayer or pseudostratified columnar epithelium, mainly composed of closely arranged columnar epithelial cells, mucous cells, and digestive cells [19,51,53]. The loose inner connective tissue, which is mesodermally derived [52], contains collagen fibers, undifferentiated cells, lymphocytes, amoebocytes, and morula cells [51]. The muscle layer, which is also mesodermally derived [52], is arranged in two different directions. The phagocytic coelomocytes are localized in the basement membrane of the coelomic epithelium (also called peritonaeum), and their long, narrow, basal projections extend through the muscle layers and nerves to the connective tissue [53]. Coelomic epithelium is mainly composed of peritoneocytes and myoepithelial cells [51,54] with adjacent outer connective tissue. Some researchers refer to the coelomic epithelium, outer connective tissue, and muscle layer together as the mesothelium [55]. Recent studies suggest that there are three main tissue layers: the mesothelium, a connective tissue layer, and the luminal epithelium [23].

The digestive tract is anchored to the body wall by a continuous mesenteric network and suspended in the body cavity. According to the position of each part of the mesentery, it is referred to as dorsal, ventral, or lateral [19]. The structure of the mesentery is mesothelial layer–connective tissue layer–mesothelial layer, which means that the mesentery contains all of the histological components that make up the digestive tract, except for the luminal epithelium [19,56].

### 3.1. Wound Healing Stage

Ruptures of the esophagus, stomach, and cloaca heal into a blind end, and the edge of the mesentery is repaired rapidly to prepare for the formation of a new intestine [27]. In the anterior evisceration holothurians, the wound at the front of the body is closed due to the contraction of the body wall, where radial longitudinal muscles and water canals are converged [33,34]. During this process, a disordered muscle layer can be observed, and the myofilaments condense into a compact spindle-like structure (SLS), which is a sign of muscle dedifferentiation [27,57,58,59,60,61]. At this stage, the number of juvenile cells thought to be the stem cells of coelomocytes increases and differentiates into amoebocytes and morula cells, and the amoebocytes perform phagocytic functions [34,57,61,62]. Morula cells are involved in the cutaneous wound healing process, but not in intestinal regeneration, as the number decreases at the initial stage of intestinal regeneration [63,64]. SLSs are accumulated in the cytoplasm and subsequently expelled outside the cell to be phagocytosed by amoebocytes [65], thus, cells with SLSs present in the cytoplasm are not necessarily dedifferentiated muscle cells. Peritoneal cells, the other major cell type of the mesothelium, also dedifferentiate at this stage, with fascicular intermediate filaments splitting into short fragments [34]. 

Wounds are responsible for the stress condition, which induces cellular and biochemical responses. Vazzana et al. [66] reported increased expression of heat shock protein 70, which is expressed in coelomocytes, and speculated that the function is to induce cell differentiation and migration to the fracture plane. 

### 3.2. Anlage Formation Stage

Intestinal regeneration begins at the free end of the mesentery [19,48,55,56,57]. Initially, differentiated cells in the mesothelial layers dedifferentiate and transdifferentiate in preparation for migration to the free end of the mesentery, where regeneration starts [54,56,67]. There is no direct evidence that stem cells in the traditional sense are involved in regeneration [54,68]. Apart from the thickening of the mesenteric margin, two anlages [34] form at the esophagus and cloaca in posterior eviscerating species such as *A. japonicus* [69] and *H. glaberrima* [48], whereas the anterior anlage consists of a mass of cells and posterior anlage, namely a blind tube, in anterior eviscerating species such as *Eupentacta fraudatrix* [34] and *E. quinquesemita* [33].

The content of the extracellular matrix changes significantly during anlage formation. The content of fibrocollagen and collagen decreases in a gradient from the free end of the mesentery to the body wall, which is related to the proteolytic activity of matrix metalloproteinases [64,70], and the same pattern occurs for myocytes [23]. At this stage, a great deal of myocytes dedifferentiation and migration occurs. During migration, the cells lose their dedifferentiation characteristics, and tonofilaments and spindle-like structures disappear from the cytoplasm [34,54]. Dedifferentiation of myocytes might result in cell proliferation [57], but only a small amount of cell proliferation occurs at this stage; the level increases in the next stage [48]. This has been demonstrated in several species, including *H. glaberrima*, *Stichopus mollis*, and *E.*
*fraudatrix* [64]. García-Arrarás et al. [67] reported that some epithelial cells sank into connective tissue and became mesenchymal cells at 3 days post-evisceration in *H. glaberrima*. In another study, Okada and Kondo [33] found mesenchymal cells in the anterior mesentery margin and during the mesenchymal–epithelial transition (MET) in *E. quinquesemita*, but there is no MET in *E. fraudatrix* because this species retains intercellular junctions [34,71]. In the next step, dedifferentiated cells proliferate, thereby providing cells for the intestinal rudiment. Apoptosis, which is activated by injury or remodeling, is a cytological event accompanied by cell proliferation and may be an inducer of cell proliferation [23,72].

### 3.3. Lumen Formation Stage

The intestinal lumen is formed in opposite directions from the anterior and posterior ends and completely connects into a thin and fragile intestinal tube. Regardless of the anterior evisceration or the posterior evisceration, the luminal epithelium of the posterior anlage originates from the endoderm-derived lining epithelium of the cloacal [34,48]. In the anterior anlage, there are two variants. For the posterior evisceration sea cucumbers, represented by *H. glaberrima* and *A. japonicus*, the esophagus remains in the coelom after evisceration [27,37,48]. The new intestinal lumen is formed by the invasion of tubular residues of the esophagus into the amorphous matrix of the thickened connective tissue [27]. For the anterior evisceration sea cucumbers, represented by *E. quinquesemita* and *E. fraudatrix*, the entire gut and oral complex are excluded except the cloaca [33,34]. The anterior mesenteric margin thickens to form the anlage filled with mesenchymal cells. Mesenchymal cells differentiate into epithelial-like cells, and multiple cavities are formed [33].

The lumen epithelium is derived from the proliferation and differentiation of cells in the esophageal mesothelium, and the proliferation of epithelial cells forms the intestinal lumen [40,48,73]. During this stage, cell division is most active in the luminal epithelium and is evenly distributed [27], with a gradual increase in the coelomic epithelium and muscular layers.

### 3.4. Intestinal Differentiation Stage

This stage is dominated by the proportioning of tissue layers and differentiation of different regions of the intestine. The intestinal tract develops digestive function and forms a typical S-shaped structure. Once the intestinal cavity is formed, the new intestine develops from a rudiment with only two tissue layers into a new intestine with three tissue layers [23,74]. The differentiated cells in the luminal epithelium undergo cell differentiation and proliferation so that the intestine gradually becomes a multicellular layer of tissue.

### 3.5. Intestinal Growth Stage

After intestinal differentiation, the digestive tract is structurally and cellularly indistinguishable from the normal intestine, but it is smaller in size than the normal intestine. The tissue layers of the intestinal wall thicken further, eventually returning to the size of the intestine before evisceration. It takes about 1 month for the regenerative intestines of *A. japonicus* and *H. glaberrima* to return to the same morphology and function as normal intestines.

## 4. Molecular Mechanism of Intestinal Regeneration

### 4.1. Intestinal Regeneration

Studies of the molecular mechanisms involved in intestinal regeneration in sea cucumbers began around 2000 [27], so little is known and much remains unknown. Prior to the development of sequencing technology, studies of the molecular mechanisms of sea cucumber regeneration focused on functional exploration of individual genes (i.e., regeneration-related or developmentally relevant genes that have been studied in other animals) [27]. In situ hybridization or Northern blot and immunohistochemistry are used to localize, quantify, and characterize the transcripts and translated proteins of candidate genes to determine when and where the genes are expressed. Santiago et al. [75] cloned the serum amyloid A (SAA) protein gene from the cDNA library of the regenerating intestine of *H. glaberrima* [76] and found that its expression increased during the regeneration process, reaching a peak at 15 days post-evisceration. In another study, Santiago-Cardona et al. [77] reported that lipopolysaccharides could induce an increase in SAA mRNA levels in non-regenerating intestines. Suárez-Castillo et al. [78] and Zheng et al. [79] cloned the Ependymin gene from regenerating libraries of *H. glaberrima* and *A. japonicus*, respectively, and found that it was overexpressed during intestinal regeneration. Mashanov et al. [72] reported that two pro-cancer genes, *survivin* and *mortalin*, were overexpressed only during regeneration in the sea cucumber, and Mashanov et al. [80] found that *Wnt9*, *TCTP*, and *Bmp1/Tll*, which have been previously known to be implicated in embryogenesis and cancer, were up-regulated in visceral regeneration in the sea cucumber *H. glaberrima*. In addition to cloning genes from regenerating libraries, researchers also obtained genes by rapid amplification of cDNA ends polymerase chain reaction, such as *Wnt6* and *Hox6* [81], elongator protein 2 (*Ajelp2*) [82], *WntA* [83], and Krüppel-like factor13 (*klf13*) [74].

Adams et al. [84] first proposed the concept of expressed sequence tags (ESTs), which is a classical technique complementary to other methodologies such as nucleic acid microarray and serial analysis of gene expression. Zheng et al. [79] and Rojas-Cartagena et al. [85] successively constructed the cDNA libraries of the early regenerating intestine and the normal intestine of *A. japonicus* and *H. glaberrima*, and they analyzed the ESTs from these two libraries for each species. For the first time, the functions of differentially expressed genes were outlined, with decreased expression of genes involved in metabolism and increased expression of genes involved in immunity, division, and cell signaling transduction in the regenerating intestine [79]. Subsequently, Ortiz-Pineda et al. [52] constructed the gene expression profile based on the EST database for the anlage and lumen formation stages. With the development of high-throughput sequencing technology, Sun et al. [86] constructed a library containing 182,473 reads using 454 cDNA sequencing technology. Transcriptome sequencing technology overcomes the shortcomings of microarray technology, such as low throughput, high background signal interference, and low sensitivity, and it is an effective tool for constructing high-quality expression profiles. Sun et al. [87] constructed gene expression profiles of *A. japonicus* in different regeneration stages and used RNA-seq technology to reveal the dynamic changes in gene expression that occur during regeneration. The gene expression profiles mentioned above were based on transcriptome data without a reference genome because the first whole genome sequence of the sea cucumber was not constructed until 2017 [88]. Through scaffold splicing and amino acid sequences analysis, Zhang et al. [88] found two gene families: Prostatic secretory protein of 94 amino acids (PSP94)-like genes, which have 11 tandem duplications, were overexpressed during the wound healing stage; and fibrinogen-related protein genes, which have 21 tandem duplications, were significantly up-regulated during the early and middle stages of visceral regeneration. The whole genome of *H. glaberrima* has also been constructed and the melanotransferrin (Mtf) gene family was found to be significantly expanded in this sea cucumber [89]. Fumagalli et al. [2] mapped transcriptomic data from different stages of the hydra *Hydra magnipapillata*, the flatworm *Schmidtea mediterranea*, and *A. japonicus* to the same reference database and screened for some regenerative conserved genes. These three species share 18 regenerative differentially expressed genes with humans, and their functions involve “proteases involved in developmental processes”, “calcium ion binding”, “cell junction”, “structure and migration”, and “cancer-related scramblase ANO7”.

Intestinal regeneration involves the coordinated regulation of multiple genes, so in addition to focusing on the function of individual genes, researchers have explored related signaling pathways and gene families. The Wnt signaling pathway plays a role in the differential expression of genes during intestinal regeneration via three upstream genes (*Wnt7*, *Fz7*, *Dvl*) and one downstream gene (*c-Myc*), which are conserved and highly expressed [90]. The retinoic acid signaling pathway can regulate cell division and dedifferentiation [91]. The main gene families involved in regeneration are *Wnt*, *Frizzled* [68,92], *Hox* [76,88], and *Mtf* [89]. Wnt/β-catenin is a canonical signaling pathway associated with development and regeneration and can promote regeneration in axolotl, *Xenopus*, and zebrafish [93]. Hox genes are involved in creating positional information in polychaetes and planarians [94]. We have selected some genes, which are studied for their expression patterns by immunohistochemistry and listed in Table 2, which suggested that intestinal regeneration is a multi-gene-regulated process.

In addition to the spatiotemporal expression of genes, researchers have also focused on the function of gene products. Matrix metalloproteinase inhibitors cause disruption [70] or delay [99] of intestinal regeneration processes. Viera-Vera and García-Arrarás [91] reported that an inhibitor (citral) of the enzyme synthesizing retinoic acid reduced the size of the regenerated intestine. Additionally, a Wnt pathway inhibitor (iCRT14) was found to block cell dedifferentiation and reduce the size of the regenerated intestine, whereas a Wnt pathway activator (LiCl) increased its size [100]. RNA interference, which identifies specific gene sequences and their functional roles, is also being applied to echinoderms. Yuan et al. [90] reported that down-regulation of *Wnt7* and *Dvl* expression by RNA interference significantly inhibited intestinal elongation. An in vitro primary cell culture system for sea cucumbers has been established [101], and it provides support for modern molecular biology techniques such as gene transfer and single cell transcriptomics.

### 4.2. Neural Components Regeneration

The enteric nervous system is arranged in plexuses and is distributed in all tissue layers of the mesentery and intestine. These plexuses are interconnected networks of neurons, axons, and enteric glial cells [102]. The mesenteric mesothelium has extensive fiber tracts and associated neurons that extend from the body wall to the intestine. Neuron-like cells and fibers are also present in the connective tissue, but the abundance and presence are different from the mesothelial layer with small amounts of nerve fibers that separate from each other. Nerve fibers and cells in mesentery rarely undergo morphological changes during regeneration, with only mild disturbance of the mesothelial plexus and degradation of fibers from the free end of the mesentery [55]. Neurodegenerative processes are also observed in the anlage, and the degradation is not completed until 10 dpe. Onset of innervation is observed in the mesothelial layer only after intestinal lumen formation (14 dpe). After that, nerve fibers in the connective tissue projected from mesothelial layer are observed. After the formation of luminal epithelium, neurons and neuroendocrine cells appear [103]. Consequently, regeneration of neural components lags behind intestinal regeneration.

## 5. Conclusions

Sea cucumber intestinal regeneration has been studied for many years. In this article, we reviewed the process of autotomy, evisceration, and regeneration of sea cucumbers, and described the cytological changes. With the development of high-throughput sequencing technology, the genomes of multiple sea cucumber species have been assembled, which greatly promoted the progress of molecular research on intestinal regeneration. However, there are still many unsolved mysteries. What substance causes the sea cucumber to eviscerate, neuropeptide or hormone? Is the regenerative anlage formed just by cell dedifferentiation and transdifferentiation, and are there stem cells in the body? By what mechanism does the sea cucumber initiate regeneration?

From an evolutionary perspective, compared to other regeneration model species, such as planarians, zebrafish, and amphibians, echinoderms are still left behind. Single-cell transcriptomes have been applied to the regeneration of hydra [104], planaria [105], amphibian [106], and zebrafish [107], and cell types important for regeneration have been discovered. In the future, single-cell transcriptome and spatial transcriptome will help reveal typical cell type and spatial orientation. Thus, we can analyze the conserved molecular mechanism regulating regeneration process among achordate and chordate.

## Figures and Tables

**Figure 1 biology-11-01095-f001:**
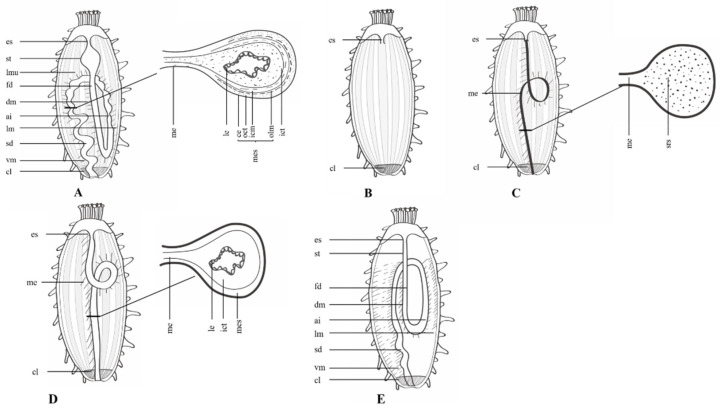
Diagram of the intestinal regeneration pattern of sea cucumbers. (**A**) Structure of the normal sea cucumber digestive tract. (**B**) After evisceration. (**C**) Thickening of the mesentery and formation of a solid rod-shaped structure. (**D**) Lumen formation. (**E**) Differentiation and growth of the newborn intestine. *Abbreviations*: es, esophagus; st, stomach; fd, first descending intestine; dm, dorsal mesentery; lmu, longitudinal muscle; ai, ascending intestine; lm, lateral mesentery; sd, second descending intestine; vm, ventral mesentery; cl, cloaca; me, mesentery; ce, coelomic epithelium; oct, outer connective tissue; olm, outer longitudinal muscles; icm, inner circular muscles; mes, mesothelium; ict, inner connective tissue; le, luminal epithelium; srs, solid rod-shaped structure.

**Table 1 biology-11-01095-t001:** Patterns of Evisceration and Regeneration.

Species	Order	Eviscerating			Regenerating		Reference
		Autotomy Site	Evisceration Site	Excreted Viscera	Initial Regeneration Site	Regeneration Cycle	
*Eupentacta quinquesemita*	Dendrochirotida	(1) the oral body wall/the oral complex (2) the longitudinal muscle/the retractor muscle (3) intestine/mesentery (4) intestine/cloaca	rupture of anterior end of the body (anterior)	the oral complex; the digestive tract; part of the gonads	the free end of the anterior mesentery	2–3 weeks	[33]
*Eupentacta fraudatrix*		(1) rupture of the introvert (2) the pharyngeal retractor muscles/the pharyngeal retractor muscles (3) intestine/cloaca (4) intestine/mesentery		the entire digestive tube; the oral complex	free end of the dorsal mesentery at the healed oral end of the body; the free end of the ventral mesentery at the cloaca	27 days	[34]
*Sclerodactyla briareus*		(1) the introvert and the muscular body wall (2) the intestine/the cloaca (3) intestine/mesentery		stomach; intestine; oral complex	the free end of the remaining mesentery	20–37 days	[35,36]
*Holothuria glaberrima*	Holothuriida Synallactida	(1) the esophagus/the descending small intestine (2) the large intestine/the cloaca (3) intestine/mesentery	cloaca (posterior)	the intestinal system; the hemal system; the right respiratory tree; most of the gonads		28 days	[23]
*Apostichopus japonicus*		(1) the esophagus/the stomach (2) the intestine/the cloaca (3) intestine/mesentery		the intestinal system (except esophagus)	the free end of the remaining mesentery; anlage arise from the esophagus; anlage arise from the cloaca	21 days	[37]
*Holothuria polii*		(1) the esophagus/the stomach (2) the intestine/the cloaca (3) intestine/mesentery		the intestinal system (except pharynx and esophagus); the left respiratory tree; gonads		6 weeks	[38]

**Table 2 biology-11-01095-t002:** Single gene expression patterns during intestinal regeneration.

Genes Type	Gene Name	Gene Expression During Regeneration	Species	Expression Site	Putative Function	Process Involved in	Reference
MMPs	Ef-72 kDa type IV collagenase	-	*Eupentacta fraudatrix*	coelomic and luminal epithelia	degradation of ECM proteins and facilitating cell movement	ECM remodeling and cell migration	[95]
	Ef-MMP16	-	*Eupentacta fraudatrix*	coelomic epithelium	migration and/or proliferation of coelomic epithelial cells	migration of coelomic epithelial cells	[95]
	ajMMP-16 like	high expression during the whole process, with the highest expression at 1 dpe followed by a constant drop to normal level from 7 dpe to 21 dpe	*Apostichopus japonicus*	no expression in normal intestine; in the regenerative intestine, expressed at coelomic and luminal epithelia	degrading ECM and growth factors; targeting ECM components and biological molecules	ECM remodeling	[96]
	ajMMP-2 like	highest expression at 6 hpe, slightly decreasing to approximately 2 at 1 dpe and 3 dpe, gradually declining to normal level	*Apostichopus japonicus*	no expression in normal intestine; in the regenerative intestine, expressed at luminal epithelia			[96]
TIMPs	Ef-tensilin3	-	*Eupentacta fraudatrix*	coelomic epithelium and the ventral part of the luminal epithelium; opposite to EF-MMP16	inhibiting the activity of MMPs	ECM remodeling	[95]
immune-related gene	Serum amyloid A	high expression during the whole process, with the highest expression at 15 dpe	*Holothuria glaberrima*	coelomic epithelium	stimulate cell migration and adhesion to an ECM substrate; involved in the formation of luminal epithelium and muscular layer	ECM remodeling; formation of luminal epithelium and the muscular layers	[75]
Melanotransferrin (MTf)	Hg MTf1/Aj MTf	increased expression from day 3, reaching the peak at 7 dpe	*Holothuria glaberrima*	mesothelium	immune activation	dedifferentiation of mesothelial cells and the following proliferation and migration	[97]
	Hg MTf2						
	Hg MTf3	high expression at 3 dpe and 5 dpe, followed by decreasing gradually to normal level					
	Hg MTf4						
Wnt signaling pathway	WntA	up-regulated significantly from 6 h to day 14 with the maximum expression at 14 dpe	*Apostichopus japonicus*	in the luminal epidermal, muscle layer, and submucosa	be connected with cell proliferation	the proliferation, dedifferentiation and migration of luminal epithelium, muscle layer, and submucosa cells and apoptosis in the basal lamina of the mucosal epithelium at each stage of wound healing	[83]
	Wnt9	strongest expression on days 7–14	*Holothuria glaberrima*	in normal intestine, no expression; in the regenerative intestine, expressed at mesothelium	control the transitions between the dedifferentiated mesothelial cells and the mesenchyma	a series of activities of the mesothelium	[80]
gene involved in embryogenesis	Bmp1/Tll	up-regulated on days 3 through 12	*Holothuria glaberrima*	in the apices of the developing folds of the luminal epithelial	remodeling of extracellular matrix	morphogenetic movements leading to folding of the luminal epithelium and gut looping	[80]
	TCTP	highest expression at 3 dpe and 21 dpe	*Holothuria glaberrima*	in normal intestine, expressed at the apices of the luminal epithelial; in the regenerative intestine, expressed at mesothelium similar to survivin and mortalin	apoptosis suppression and regulation cell proliferation	deep transient dedifferentiation of mesothelial cells	[80]
	survivin	insignificant increase at 7 dpe and 14 dpe, significant increase at 21 dpe	*Holothuria glaberrima*	in normal intestine, expressed at the base of the luminal epithelium; in the regenerative intestine, mostly expressed at mesothelium	cell proliferation; suppression of the programmed cell death	dedifferentiation of mesothelial cells and extensive proliferation	[72]
	mortalin	two peaks of roughly 3-fold up-regulation at 7 dpe and 21 dpe	*Holothuria glaberrima*	in normal and regenerative intestine, expressed at mesothelium	cell proliferation		[72]
	piwi	high expression during the whole process, with the highest expression at 3 dpe	*Eupentacta fraudatrix*	in normal intestine, expressed at mesothelium; in the regenerative intestine, expressed at ECM	-	do not participate in the formation of the luminal epithelium	[95]
Sox gene family	Ef-Sox9/10	-	*Eupentacta fraudatrix*	coelomic epithelium, mesenchymal cells, and the developing luminal epithelium	regulating the differentiation of mesenchymal cells into epithelial cells	coelomic epithelium transdifferentiation; the redifferentiation of myoepithelial cells and the formation of muscle layer	[95]
	Ef-Sox17	-	*Eupentacta fraudatrix*	at the site of immersion only in surface cells	regulation of the initial stages of transdifferentiation		[95]
pluripotency factors (Yamanaka factors)	SoxB1	down-regulation at the early post-evisceration stages (3 dpe and 7 dpe)	*Holothuria glaberrima*	in normal intestine, in the luminal epithelium; no signal at 3 dpe; expressed in the mesothelium at 7 dpe; finally expressed in the luminal epithelium	not essential for cell dedifferentiation	-	[98]
	Myc	Up-regulated at 3 dpe, but then returned to normal level, before being slightly (∼1.5 fold) down-regulated at 21 dpe		in normal intestine, in the luminal epithelium and mesothelium; expressed in the mesothelium at 3 dpe; expressed in the mesothelial epithelial cells at 7 dpe; finally expressed in the luminal epithelium	play an important role in dedifferentiation/regeneration	dedifferentiation of mesothelium	[98]
	Klf1/2/4	no significant differences		in normal intestine, in the luminal epithelium; expressed in the mesothelium at 3 dpe; expressed in the mesothelial epithelial cells at 7 dpe; finally expressed in the luminal epithelium	-		[98]
	Oct 1/2/11				-		[98]

## Data Availability

Not applicable.

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
