# Peer review of "A Review of Histocytological Events and Molecular Mechanisms Involved in Intestine Regeneration in Holothurians"

_biology, 2022, doi:10.3390/biology11081095_

Round 1

Reviewer 1 Report

The authors have addressed most of the issues presented after an initial review of the manuscript.  I am sorry to say that there are many remaining issues that need to be addressed for the manuscript to be presentable to a wider audience.  Many of these are minor issues, but they add up and should be fixed prior to publication.

Line 21-  What is meant by “nodal”.?  Is this the correct word?

Line 42- fix “parts, they”

Line 47-49-  To say that mammals rely on undifferentiated stem cells is a very simplistic generalization of a very complicated topic.  I am sure that many regenerative biologist would be opposed to calling dedifferentiated or transdifferentiated cells as “stem cells”.  This is because cells labeled as “stem cells” have very typical characteristics, one of them being the ability to form new stem cells.

Line 67- very awkward saying “…for the first time (Ref)”- which implies that it was published, and then following the sentence with “and published the research results (another ref)”

Line 69-73-  the authors need to be careful with such statements as “ returns to normal physiological state”. Or “returns to its original shape”.  -  Many of the description made in the original articles are very superficial and few, if any, functional experiments have been done to show that the regenerated intestine is similar in function to the original one.  Most of what has been published is assumed from the finding that the new intestine has some intestinal content in its lumen.

Line 87- Excretion in scientific terms implies “eliminating metabolic waste”. Maybe use a different verb.

Line 127 – This sentence is completely out of place.  First, it is not fully related to the topic of the paragraph.  And even if it were, transcriptome studies cannot confirm that a process consumes energy.

Line 128-  not sure what is meant by “collagen structures”

Line 134- in homogeneous…..structures…..lysosomes.

Line 150-168 -  To the knowledge of this reviewer, the so-called “evisceration factor” has not been characterized, even though the A japonicus genome is available.  Most of what has been published about it, refers to an unknown and putative substance.  This has to be explained in the text, particularly when stating that EF is a “multifunctional molecule”. Or stating that it might be a “neuropeptide”. Or saying that “KCl stimulates the release for acetylcholine and EF”.

If this has been shown, then the correct references must be included.

Figure 1- This figure needs to be improved.  First, the order of the drawings is not appropriate.  If it is read from left to right it goes a-c-b… If it is read from top to bottom then it is a,b,d.   Then there are descriptions in A but not in the others.

Line 198 and 209, 210-  the description of an “outer connective tissue” layer that appeared in the early literature of the holothurians, has disappeared from more recent works, particularly those using EM where the fine details are clearly seen.

Line 201 and 217- repetition

Line 237-238-  This manuscript does show a differential expression of HSP 70, however it does not show any functional evidence. 

Line 242- is there evidence for dedifferentiation of cells in the connective tissue layer.

Line 245-249. The text is very confusing.  Particularly where it says that “ whereas the anterior anlage consists of a mass of cells…” sort of implying that the other anlages do not consist of a mass of cells.

Line 274-  Saying that “the esophagus residues remained” is wrong.  These are not remains…the complete esophagus can still be found.

Line 278-  What does “without residue retention” means?

Line 292- The authors do not explain anywhere that the luminal cells dedifferentiate.

Line 300-  What does “return to normal” means?

Line 319-321-  The manuscript by Mashanov et al. shows the presence of these genes. mRNA but the functional aspects are inferred.  No experiment was done to show that the genes are doing what is mentioned here.

Line 355-356-  Are these regeneration stages?

Line 364-  The Wnt pathway plays a role in the differential expression of genes….

Line 373-374-  This sentence is confusing and not accurate.  What are the authors calling “stem cell factors”?  Many of the genes that are mentioned were originally found associated to developmental processes and, while they might be associated with some stem cell functions, they are by no way exclusive "stem cell factors".

Author Response

We would like to thank you for your careful reading, helpful comments, and constructive suggestions, which has significantly improved the presentation of our manuscript.
We have carefully considered all comments from the reviewers and revised our manuscript accordingly. The typos and grammar errors we found have been corrected.
We have uploaded a  point-by-point response as a Word file. Please see the attachment. We hope our revised manuscript can be accepted for publication.
Thank you again!

Reviewer 2 Report

There are some grammatical errors

1. Line 42

replace “partsthey” with “parts they

2. Line 81 and Table 1

replace “Holothuriids, Synallactids” with “Holothuriida, Synallactida

3. Line 103

replace “Dendrochirotids” with “dendrochirotids”

4. Line 106

replace “Synallactids” with “synallactids

5. Line 116

replace “Holothuriids, Synallactids” with “holothuriids, synallactids

6. Line 417

remove one “been”

Author Response

We would like to thank you for your careful reading, helpful comments, and constructive suggestions, which has significantly improved the presentation of our manuscript.
We have carefully considered all comments from the reviewers and revised our manuscript accordingly. The typos and grammar errors we found have been corrected.
We hope our revised manuscript can be accepted for publication.
Thank you again!

Response to Reviewer 2 Comments

Point 1: Line 42

replace “partsthey” with “parts they”

Response 1: Thank you for your suggestion. We have revised. Thanks again!

Point 2: Line 81 and Table 1

replace “Holothuriids, Synallactids” with “Holothuriida, Synallactida”

Response 2: Thank you for your suggestion. We have revised. Thanks again!

Point 3: Line 103

replace “Dendrochirotids” with “dendrochirotids”

Response 3: Thank you for your suggestion. We have revised. Thanks again!

Point 4: Line 106

replace “Synallactids” with “synallactids”

Response 4: Thank you for your suggestion. We have revised. Thanks again!

Point 5: Line 116

replace “Holothuriids, Synallactids” with “holothuriids, synallactids”

Response 5: Thank you for your suggestion. We have revised. Thanks again!

Point 6: Line 417

remove one “been”

Response 6: Thank you for your suggestion. We have revised. Thanks again!

Reviewer 3 Report

The authors have addressed my questions and suggestions and have improved the manuscript.

Author Response

We would like to thank you for the time and effort spent in reviewing the manuscript. We have carefully considered all comments from the reviewers and revised our manuscript accordingly. The typos and grammar errors we found have been corrected.
Thank you again!

Round 2

Reviewer 1 Report

The authors have addressed most of my concerns

This manuscript is a resubmission of an earlier submission. The following is a list of the peer review reports and author responses from that submission.

Round 1

Reviewer 1 Report

Su and colleagues have submitted a review manuscript entitled “A review of histocytological events and molecular mechanisms involved in intestine regeneration in holothurians”.

In this review the authors attempt to put together many areas of holothurian intestinal regeneration but in trying to cover a large number of areas (autotomy, anatomy, histology, regenerative stages, genes, techniques and regeneration mechanism, among others) they only provide a superficial analysis on them.  Take for example the comparisons made in Table 1.

Although the authors present what has been published in different species, there is little integration.  Where they state the initial regeneration site, various terms are used that makes comprehension difficult- edge of the anterior mesentery vs free edge of dorsal mesentery vs free margin of the mesentery vs free end of the remaining mesentery.  Are all these sites equivalent?  What is the relation among them?  The authors bypass the chance of clearly explaining what is known to be common or what differs among the various species that have been studied.

Similarly, on the paragraph where they describe the intestine organization and histology, these are compared to the mammal/vertebrate intestine instead of to one another.  Different terms are used without stating if they identify different cells/structures in one species, one compartment of the digestive tract or if they are a common element to all holothurians.  For example, coelomic epithelium vs peritoneal epithelium vs mesothelium vs peritonaeum.  Or when defining cells such as deformable cells, mulberry cells, myoepithelial cells or phagocytic coelomocytes.

A bit troubling is the tendency of the authors to extrapolate results among species or to present an incomplete statement that could provide the base for a wrong conclusion.

See the following examples:

Line 48- “mammals only have the ability to regenerate locally using the reserve of stem cells in vivo”.  However liver regeneration, which is common in mammals does not occur by stem cells.

 Line 61- “sea urchins have limited regenerative capacity, and they are only able to regenerate tube feet and spine”.  Sea urchin can also regenerate pedicellariae and their calcium skeleton or test.

Line 66- “The literature about regeneration by sea cucumbers can be traced back to 1956”.  In the previous line they state that Bertollini did experiments in the 1930’s.  These experiments were published and are the first record of the phenomenon in the scientific literature.

Line 110-116-  the authors discuss “another mechanism of autotomy” but only mention its occurrence in two holothurian orders, leaving the readers in the dark of whether these are the only orders that can do it or whether they just wanted to highlight them.

Line 231- “These cells are considered to be amoebocytes, which act as scavengers to phagocytose cellular debris and deposits of extracellular matrix proteins”. Maybe the authors need to reconsider the term amoebocyte which has been used quite indiscriminately by multiple authors to define some cell types.  Or are they implying that muscle cells become amoebocytes.

Line 238- “During the wound healing process, fewer inflammatory cytokines results in faster wound healing [66].”  Is this true in echinoderms?  Has it been shown?

The section on molecular mechanisms of intestinal regeneration is mainly a listing of techniques that have been used to identify putative or candidate regeneration-related genes.  The authors need to clearly relate the message that the large majority of these studies only provide a list of genes whose expression correlates with some regenerative process but that there is no proof that the gene is actually involved in the process. Therefore, the function of the genes in the regenerative process cannot be determined.

The authors present a Table (#2) where some of these genes are listed. However, there is no clear explanation why these (and not other) genes were chosen.  Are these genes that have been identified in more than one species?

The authors erroneously label the first column of their Table as “Cells” where it should be “Genes”

Finally- the authors should check their use of references since in many cases they seem either to lack the correct reference or these are displaced.  Just one example.  Dolmatov (#37) is used to back up the role of the nervous system during evisceration (not adequate) but is not used to explain autotomy by fission where it should have been prominent.

In summary, I do not believe that the manuscript, as presented, is ready for publication.  My recommendation is for the authors to focus on some of the topics and integrate what is known among the various holothurians models that they have analyzed to clearly present a comparative analyses of cells, mechanism or structures that the animals have in common and those where they differ among themselves.

Author Response

Please find the reply in the attachment.

Reviewer 2 Report

This is a very interesting review about the current state of research concerning the mechanisms involved in regeneration of the intestines in holothurians. The subject has not received as much exposure as regeneration research in traditional regeneration models (newts, fish, planarians, hydra, to name a few). Because the mechanisms involved are certain to both differ but also to convene on common ideas between taxons, it is important to re-adjust one’s views of this important field.

There are a number of issues that this reviewer thinks would improve the manuscript.

  • The review needs a simple cartoon, accompanying figure 1, that depicts the body plan of a typical holothurian and which serves as a reference to the terms in the columns “autotomy site”, “excreted viscera” and initial regeneration site” in table 1.
  • Section 2 would benefit a little more elaboration on possible ideas in which physiological context nervous control might be able to trigger/promote evisceration. Could this be the release of factors acting like hormones or like local stimuli, or is muscle activity involved?
  • From an evolutionary point of view, I would find it interesting to learn a little more about the evolutionary cost and benefits of autotomy and evisceration, particularly when this happens not in response to impending predation, but is a recurring/seasonal event.
  • Line 106 hints at exceptions in the ability of some species to eviscerate anteriorly. Along this line, it would be useful to also refer a reference where important traits and their differences in holothurian regeneration have been mapped on a phylogenetic tree.
  • In general, much of the review, particularly in section 4, reads like a list of loosely ordered findings. It would be very useful if the authors could summarize these findings and put them in context to what is known to organ regeneration processes in zebrafish and amphibians.
  • The review ends somehow suddenly without something of an outlook or synthesis of the authors own thought about where the field is headed, what the most important questions are and which approaches currently hold the most promise to reveal how regeneration processes are guided by molecular processes. This deficit should be mended.
  • There is a missing full stop after “stages” in line 333.

Author Response

(The authors gave the same response as above.)

Reviewer 3 Report

Manuscript of Fang Su, Hongsheng Yang  and Lina Sun “A review of histocytological events and molecular mechanisms involved in intestine regeneration in holothurians” is devoted to the review of mechanisms of intestinal regeneration in sea cucumbers. This manuscript is well written and easy to follow. Its value lies in the analysis of a large number of publications devoted to various aspects of sea cucumber regeneration. Nevertheless, the manuscript needs serious editing and cannot be published in this form.

Part 2. Autotomy and evisceration.

The authors point out that “...sea cucumber species of the orders Aspidochirotida, Dendrochirotida, and Apodida are capable of eviscerating their intestine...” (lines 79-82).

First, apodids cannot eviscerate their viscera. They are capable of autotomy of the posterior part of the body, as a lizard separates its tail. Secondly, according to modern taxonomy, there is no order Aspidochirotida. There are orders Holothuriida and Synallactida (see Miller A. K., Kerr A. M., Paulay G., Reich M., Wilson N. G., Carvajal J. I., Rouse G. W. Molecular phylogeny of extant Holothuroidea (Echinodermata). Mol Phylogenet Evol. 2017. Vol. 111. P. 110-131.)

Table 1.

In Table 1, the “Evisceration site” column for dendrochirotids indicates "mouth". However, the mouth of these animals is located on the oral complex (aquapharyngeal bulb) and is removed along with the viscera. It is more correct to indicate "rupture of anterior end of the body". Lantern, oral complex, and aquapharyngeal bulb is the same. It's better to use one term, oral complex or aquapharyngeal bulb. “Aspidochirotids” should be replaced with “Holothuriids” and “Synallactids” not only in the table, but throughout the text.

Line 42

“copies” replace with “parts”

Lines 53-55

Sentence “Huet et al. [15, 16] studied...” is not understandable.

Lines 106-108

“Aspidochirotids generally eviscerate through the cloaca, although some eviscerate by tearing the body wall.” Need to add citations.

Lines 110-112

“Another mechanism of autotomy that is similar to evisceration is fission....”

Fission is asexual reproduction. The Leptosynapta does not have a fission, but an autotomy of the posterior part of the body.

Line 116

Sclerodactyla (Thyone) briareus is not capable of asexual reproduction.

Lines 121-122.

What is “oropharyngeal ring”? The mouth and pharynx are located in the oral complex and are removed during evisceration.

Line 207

“...deformable cells, and mulberry cells [52]”. What are these cell types? I cannot find them in Mashanov et al., 2004

Lines 222-225

The meaning of the sentence is not clear.

Line 226

If I understand correctly, in chapter 3.1. the authors describe wound healing after posterior evisceration. Is there any data on wound healing after anterior evisceration?

Lines 231-232

What cells are considered as amoebocytes?

Lines 253-256

“At this stage, a great deal of myocyte dedifferentiation and migration occurs, but during migration the cells lose their dedifferentiation characteristics, and tonofilaments and spindle-like structures disappear from the cytoplasm [26, 55]”

The meaning of the sentence is not clear. Maybe replace it with: “At this stage, a great deal of myocyte dedifferentiation and migration occurs. During migration the cells lose their differentiation characteristics, and tonofilaments and spindle-like structures disappear from the cytoplasm [26, 55]”?

Lines 269-279.

This paragraph contains incorrect information. In all sea cucumbers, the intestinal epithelium in the posterior anlage is formed in the same way (García-Arrarás et al., 1998; Shukalyuk, Dolmatov, 2001; Mashanov et al., 2005), and in the anterior anlage, there are 2 variants – A. japonicus (Shukalyuk, Dolmatov, 2001) + H. glaberrima (Leibson, 1992; García-Arrarás et al., 1998) and Eupentacta (Mashanov et al., 2005; Okada, Kondo, 2019). This paragraph needs to be rewritten.

Line 290

“undifferentiated” replace with “dedifferentiated”

In some cases the species name of the sea cucumber and genes names is not italicized in the text.

Author Response

(The authors gave the same response as above.)
